# Comparison of the Quality of Life of Patients with Breast or Colon Cancer with an Arm Vein Port (TIVAD) Versus a Peripherally Inserted Central Catheter (PICC)

Brent Burbridge [1] , Hyun Lim [2], Lynn Dwernychuk [3], Ha Le [4], Tehmina Asif [5], Amer Sami [3] and Shahid Ahmed [3,*]

1    Department of Medical Imaging, College of Medicine, University of Saskatchewan,
     Saskatoon, SK S7N 5A2, Canada; brent.burbridge@usask.ca
2    Department of Community Health and Epidemiology, College of Medicine, University of Saskatchewan,
     Saskatoon, SK S7N 5A2, Canada; hyun.lim@usask.ca
3    Saskatchewan Cancer Agency, Division of Oncology, College of Medicine, University of Saskatchewan,
     Saskatoon, SK S7N 5A2, Canada; Lynn.Dwernychuk@saskcancer.ca (L.D.); amer.sami@saskcancer.ca (A.S.)
4    Clinical Research Support Unit, College of Medicine, University of Saskatchewan,
     Saskatoon, SK S7N 5A2, Canada; htl842@mail.usask.ca
5    BC Cancer Agency, Abbotsford, BC V3V 1Z2, Canada; tehmina.asif@bccancer.bc.ca
*    Correspondence: shahid.ahmed@saskcancer.ca; Tel.: +1-306-655-2710

**Abstract:** Introduction: Venous access is a crucial element in chemotherapy delivery. It remains unclear whether cancer patients prefer a port to a peripherally inserted central catheter (PICC). Our study aimed to assess cancer patients' satisfaction with their venous access device and to compare the quality of life (QoL) of subjects with a PICC to those with a port. Methods: In this prospective cohort study, EORTC QLQ-C30, and a locally developed quality of life survey (QLAVD), designed to assess satisfaction with venous access devices, were administered to breast or colorectal cancer patients over a one-year period following the device insertion. Mixed effects models were used to assess changes on mean scores at different time points. Results: A total of 101 patients were recruited over a three-year period, (PICC group, $n = 50$; port group, $n = 51$). Survey response rates for months one and three were 72% and 48%, respectively. Overall, no significant differences were noted between the two groups in relation to EORTC QOL. At three months, the mean pain scores were $3.5 \pm 2.3$ for the port and $1.3 \pm 0.75$ for PICC (<0.001). The mean score for a negative effect of the venous access device on psychosocial well-being was $6.0 \pm 4.1$ for PICC and $3.0 \pm 2.7$ for the port ($p = 0.005$). Complications related to PICCs occurred in 38% patients versus 41% with a port ($p > 0.24$). Conclusions: Although subjects with a port experienced more pain during the device insertion or access for chemotherapy, it had a smaller negative impact on psychosocial scores than the PICC. No significant differences in complications rates were observed between the two devices.

**Keywords:** quality of life; venous access; peripherally inserted central venous catheter (PICC); peripherally implanted venous access port (TIVAD); complications

## 1. Introduction

Breast and colorectal cancer are the two most common cancers in North America [1]. Patients with breast or colorectal cancer often require intravenous chemotherapy [2,3]. The vesicant nature of certain chemotherapy agents and the time frame of delivery of intravenous treatment usually requires the patient to have some form of durable and reliable venous access device. In general, for a shorter duration of treatment, a peripherally inserted, central venous catheter (PICC) is an appropriate device. A PICC is inserted peripherally in the arm, with the tip of the device situated in the central venous circulation (cavoatrial junction). The PICC has external elements that protrude from the patient to be used for the infusion of medications and fluids. Alternatively, a totally implanted,

venous port system (TIVAD or port) is implanted if the treatment is expected to be for a longer duration.

Even though central venous catheters (CVCs) are commonly used in the management of breast and colorectal cancer, little data are available regarding their impact on patients' quality of life (QoL) and satisfaction with their venous access device [4]. Our study aimed to (i) assess the QoL of patients with breast or colorectal cancer who require a CVC during their treatment; (ii) compare the satisfaction and QoL of patients with a PICC to those with an arm port device (both power and non-power devices); and (iii) determine CVC complications rates. We hypothesized that patients with a port device would have a better QoL, greater satisfaction with their device, and lower complication rates compared to patients with a PICC device.

## 2. Methods

This was a prospective, cohort, pilot study to assess two quality of life and patient satisfaction assessment tools. Ethics approval for this study was obtained from the Institution's Biomedical Ethics Board.

### 2.1. Subject Selection

All subjects with newly diagnosed breast or colorectal cancers who were seen at a major provincial Cancer Center were invited to participate in the study. Subjects were enrolled between January 2016 and December 2018. Patients with a coagulopathy, an active infection, or those who were not candidates for chemotherapy were excluded. Locally, it is empirically accepted that if intravenous treatment is to be provided for less than 6 months, a PICC is an appropriate device. Alternatively, if treatment is expected to be for longer than 6 months, a port is implanted.

The referring clinicians from the Cancer Center evaluated candidates for this project based on their initial patient contact interview, physical, and laboratory assessment. Once the duration of the required intravenous chemotherapy was determined, based on the patient's disease status, a request for the desired intravenous device was sent to the Medical Imaging Department. Consent for involvement in the research project was acquired by a research nurse.

### 2.2. Central Venous Catheter Devices

All venous access devices for this project were inserted by an interventional radiologist from the Medical Imaging Department. Consent for the venous access device was acquired by the interventional radiologist involved.

The PICC implanted for this project was a 5F, dual lumen, Angiodynamics, BioFlow, Power PICC (Angiodynamics Canada Inc., Oakville, ON, Canada). The non-power injectable arm port was a 5F, single lumen, Cook Vital Mini Titanium port (Cook Medical LLC, Bloomington, IN, USA), while the power injectable device was a 6F, single lumen, Bard Titanium Slim power port (Bard Canada, Inc. Oakville, ON, Canada).

All venous access devices were implanted into the arm, between the antecubital fossa and the axilla, using ultrasound or arm venography for venous access. The vein selected for catheter insertion was preferentially the basilic vein. The preferential arm for PICC and port insertion for trial subjects was their non-dominant arm if possible. For women with breast cancer, the contralateral arm was used for device insertion. The venous access procedures were performed using a sterile technique and local anesthetic. Patients with colorectal cancer often require frequent intravenous contrast enhanced computed tomography (CT) examinations during their care; therefore, a power injectable venous access device was implanted for this group to enable the rapid injection of intravenous contrast agents for follow-up CT scans. In general, patients with early stage breast cancer do not require frequent contrast enhanced CT scans; therefore, this group of subjects received the non-power injectable port device. The current PICC, used for all subjects, was power-injectable. Technical details for these device implantation procedures have been previously described

and were followed for this project [5,6]. As per institutional guidelines, PICCs were flushed once weekly, whereas ports were flushed every 4 weeks when not in use.

### 2.3. Subject Surveys and Clinical Follow-Up

The European Organization for Research and Treatment of Cancer General Quality of Life assessment (EORTC QLQ-C30) targeted towards patients with cancer was administered along with a locally developed Quality of Life Assessment Venous Device (QLAVD) survey comprising 30 items [4] (Table S1).

The QLAVD was modeled after a survey tool created by Marcy et al. entitled, "Questionnaire for Acceptance of and Satisfaction with Implanted Central Venous Catheter (QASSIC)". The QASSIC was statistically validated and found reliable by Marcy et al. However, this was a French language survey [7]. The QLAVD was created by a local working group of oncologists, oncology nurses, and interventional radiologists using the QASSIC, after it was translated locally into English, revising some of original questions, adding several new questions, and finalizing the structure of this new survey tool. The initial deployment of the QLAVD was completed by Burbridge et al. [4] Subsequently, Liu et al. translated the QLAVD into Chinese and performed a statistical assessment of reliability and validity. Liu et al. found that the QLAVD demonstrated good content validity, had internal consistency, and had a high degree of reliability and stability [8].

The subjects for our pilot study were asked to complete both surveys, at four different times, within one month after device insertion (baseline), then at three months, six months, and twelve months after venous access device insertion. Completion of these surveys continued until the removal of their venous access device for any reason, termination of treatment with the venous access device, or subject death (whichever came first). These surveys were administered personally to each subject by a research assistant.

The subjects were followed clinically for the detection of venous access device complications during the course of their infusion therapy, as summarized from the previous literature [9–14]. Imaging for all subjects was collected in a solitary provincial Picture Archive and Communication in Medicine System (PACS). Therefore, any venous access device complication that required imaging was easily available and was reviewed.

### 2.4. Quality of Life Constructs

The EORTC QLQ-C30 comprises 30 items, 24 of which are aggregated into 9 multi-item scales: 5 functioning scales (physical, role, cognitive, emotional, and social); 3 symptom scales (fatigue, pain, and nausea and/or vomiting); and 1 global health-status scale. The remaining six single items assess symptoms of dyspnea, appetite loss, sleep disturbance, constipation, diarrhea, and financial impact. All of the scales and single-item measures range in score from 0 to 100. The questions in these constructs were scored individually and summated, and a mean score was generated. The principle for scoring the scales was the same for all patients. Firstly, the average of the items that contributed to the scale was estimated to create the raw score; then, a linear transformation was performed to standardize the raw data [15]. The mean scores were compared to assess potential differences in responses between the PICC and port groups.

A similar strategy was employed for the QLAVD survey and the questions in this survey were pooled into larger question blocks. The QLAVD survey questionnaires were aggregated into psychosocial, treatment, satisfaction, and procedural constructs. Subjects' pain experience at the time of device implantation (procedural construct) was assessed using a single question: "how painful was it for you to have your device implanted?" Subjects were asked to respond based on a scale from 1 (minimal discomfort) to 10 (severe discomfort). The psychosocial construct was calculated by summing up 18 questions relating to subjects' feelings, beliefs, and experiences of having the device implanted in relation to daily living, psychosocial, emotional, and physical activities. The higher the score, the more negative the subject's experience. The satisfaction construct consisted of three questions related to satisfaction with the venous access device. The treatment

construct was calculated by summing up 8 questions relating to the subjects' feelings, beliefs, and experiences in relation to the treatment process. The higher the score, the more negative the subject's experience.

*2.5. Statistical Analysis*

Descriptive analysis was performed to summarize the data. Mean and standard deviation estimates of each construct (procedural, psychological and treatment) were calculated separately and compared between patients with a PICC or a port for each of the four time points using Student's *t*-tests. To investigate the changes in mean scores for different constructs between the PICC and port groups across the four time points, mixed effects models were used, taking into account the interactions between the groups (type of device used) and the time points (4 survey times). Due to the high drop-out rates at times 3 and 4, mixed effects models were only run for the first two surveys (1 month and 3 month), taking into account the interaction between groups and times, controlling for other covariates such as gender, whether the subject was left- or right-handed, and the arm used for device implantation. Statistical analyses were performed with the Statistical Package for the Social Sciences (SPSS) Version 26 (SPSS Inc., Armonk, NY, USA; IBM Corp., Endicott, NY, USA). Statistical significance was set by an alpha level of 0.05.

## 3. Results

*3.1. Patients and Device Characteristics*

One hundred and one subjects were recruited: 50 in the PICC group, and 51 in the port group. The mean age of the study cohort was 55.4 years (range: 31–80 years). The dominant arm was the right arm in 98 (97%) subjects. In the PICC group, there were 29 subjects with breast cancer (men:women—1:28) and 21 subjects with colon cancer (men:women—12:9). Of the 51 patients who received a peripheral vein port, 36 had a Cook port and 15 had a Bard port. The Cook group consisted of 35 women with breast cancer and one man with colon cancer. The Bard port group consisted of 15 subjects with colon cancer: six men and nine women. In the majority of cases, ultrasound was used to access a vein.

*3.2. Survey Results*

Survey response rates for month one and month three were 72% and 48%, respectively. However, the frequency of survey responses at six months and thereafter, fell to roughly 20%. As expected, this drop-off in responses was more noticeable in the PICC group because most of these devices were planned to be in situ for less than six months. In total, 70 subjects, 36 with a PICC and 34 with a port completed both surveys within one month of insertion of the device. At three months, 48 subjects, 24 for each device, completed both surveys. Due to small number of remaining subjects at months 6 and 12 months after the insertion of the device, we are only reporting results from the first two surveys.

The mean EORTC QOL scores at one month and three months for the port and PICC groups are detailed in Table 1. Overall, no significant differences were noted between the PICC and port groups in relation to various EORTC QOL constructs at baseline and three months, except that patients with a port were found to have a significantly poorer appetite than those with a PICC at three months (*p* = 0.03).

The mean QLAVD scores for treatment-related discomfort and overall satisfaction at baseline and at three months for the port and PICC line groups are described in Table 2 (Figure 1). The mean scores for pain associated with device implantation and device utilization at baseline and at three months were significantly higher for the port group compared to the PICC line group. For example, the mean baseline pain score associated with device insertion for the port group was 5.1 (±2.9) compared to 2.8 (±2.2) for the PICC group (*p* = 0.001). Likewise, the mean baseline pain score associated with device access for therapy for the port group was 3.0 (±2.5), compared to 1.3 (±0.9) for the PICC group (*p* = 0.001). Both groups had high satisfaction scores with their devices at baseline and at three months, and overall, they felt that it was beneficial for their treatment (Table 2).

**Table 1.** Mean quality of life (QoL) scale and standard deviation (SD) at 1 month and 3 months between the peripherally inserted central catheter (PICC) line and the Port groups.

| QoL Constructs | 0–1 Months | | *p*-Value | 3 Months | | *p*-Value |
|---|---|---|---|---|---|---|
| | PICC | Port | | PICC | Port | |
| Global Health Status | 59.1 ± 21.1 | 57.1 ± 23.8 | 0.68 | 60.8 ± 22.5 | 66.7 ± 24.5 | 0.39 |
| Functional Scale | | | | | | |
| Physical functioning | 85.2 ± 13.7 | 82.8 ± 16.3 | 0.52 | 84.7 ± 13.1 | 84.4 ± 15.4 | 0.92 |
| Role functioning | 62.5 ± 35.7 | 72.7 ± 27.3 | 0.18 | 70.8 ± 33.1 | 71.7 ± 26.8 | 0.92 |
| Emotional functioning | 76.4 ± 18.7 | 71.5 ± 22.7 | 0.33 | 78.5 ± 18.0 | 79.7 ± 15.5 | 0.80 |
| Cognitive functioning | 67.1 ± 28.3 | 74.2 ± 21.7 | 0.25 | 83.3 ± 16.3 | 76.8 ± 24.0 | 0.28 |
| Social functioning | 58.8 ± 28.6 | 55.6 ± 31.1 | 0.65 | 66.7 ± 26.5 | 65.9 ± 23.3 | 0.92 |
| Symptoms Scale | | | | | | |
| Fatigue | 42.9 ± 28.3 | 44.8 ± 23.8 | 0.76 | 51.4 ± 21.9 | 40.6 ± 26.3 | 0.13 |
| Nausea and vomiting | 16.2 ± 15.2 | 13.1 ± 22.7 | 0.52 | 11.8 ± 17.4 | 4.4 ± 7.5 | 0.06 |
| Pain | 22.2 ± 27.3 | 25.3 ± 25.4 | 0.63 | 27.1 ± 29.8 | 13.0 ± 22.5 | 0.07 |
| Dyspnea | 16.7 ± 21.8 | 19.2 ± 20.5 | 0.62 | 15.3 ± 24.0 | 24.6 ± 25.1 | 0.20 |
| Insomnia | 47.2 ± 33.2 | 43.4 ± 33.8 | 0.64 | 36.1 ± 29.4 | 40.6 ± 34.8 | 0.63 |
| Appetite loss | 27.8 ± 28.2 | 21.2 ± 31.0 | 0.36 | 34.7 ± 36.1 | 14.5 ± 24.3 | 0.03 |
| Constipation | 27.8 ± 24.6 | 24.2 ± 29.2 | 0.59 | 22.2 ± 27.2 | 20.3 ± 29.7 | 0.82 |
| Diarrhea | 22.2 ± 30.9 | 34.3 ± 40.4 | 0.16 | 25.0 ± 31.5 | 18.8 ± 24.3 | 0.52 |
| Financial Difficulties | 22.2 ± 30.9 | 34.3 ± 40.4 | 0.16 | 19.4 ± 27.7 | 24.6 ± 27.0 | 0.52 |

The EORTC QLQ-C30 comprises 30 items, 24 of which are aggregated into 9 multi-item scales: 5 functioning scales (physical, role, cognitive (CF), emotional, and social); 3 symptom scales (fatigue, pain, and nausea and/or vomiting); and 1 global health-status scale. The remaining six single items assess symptoms of dyspnea, appetite loss (AP), sleep disturbance, constipation, diarrhea, and financial impact.

**Table 2.** Mean QLVAD scores and standard deviation (SD) at 1 month and 3 months for treatment-related discomfort and pain and overall satisfaction between the PICC line and the port groups.

| QLVAD Constructs | 0–1 Months | | *p*-Value | 3 Months | | *p*-Value |
|---|---|---|---|---|---|---|
| | PICC | Port | | PICC | Port | |
| How painful was it to have your device implanted? * | 2.8 ± 2.2 | 5.1 ± 2.9 | 0.001 | 2.8 ± 2.3 | 5.3 ± 2.9 | 0.002 |
| Rate the degree of discomfort you experience during needle insertion into the device * | 1.3 ± 0.9 | 3.0 ± 2.5 | 0.001 | 1.3 ± 0.75 | 3.5 ± 2.3 | <0.001 |
| Rate the degree of discomfort you experience during treatment infusion * | 1.4 ± 1.3 | 2.1 ± 1.8 | 0.10 | 1.5 ± 0.9 | 2.0 ± 1.4 | 0.12 |
| Rate the degree of discomfort you experience during needle removal after treatment * | 1.3 ± 0.9 | 1.6 ± 1.1 | 0.21 | 1.2 ± 0.5 | 1.8 ± 1.3 | 0.03 |
| Do you believe that insertion of your device was a good thing to have done? * | 8.5 ± 2.0 | 8.5 ± 2.4 | 0.92 | 9.3 ± 1.9 | 8.8 ± 1.8 | 0.36 |
| Assess your degree of satisfaction with your device * | 7.5 ± 2.3 | 8.4 ± 2.3 | 0.12 | 8.0 ± 2.3 | 8.3 ± 1.9 | 0.74 |
| If you had to have a device implanted for another session of treatment during your life, would you have another? * | 7.6 ± 3.0 | 7.8 ± 2.9 | 0.77 | 7.6 ± 3.2 | 8.3 ± 2.4 | 0.38 |

* Scale of 1 to 10.

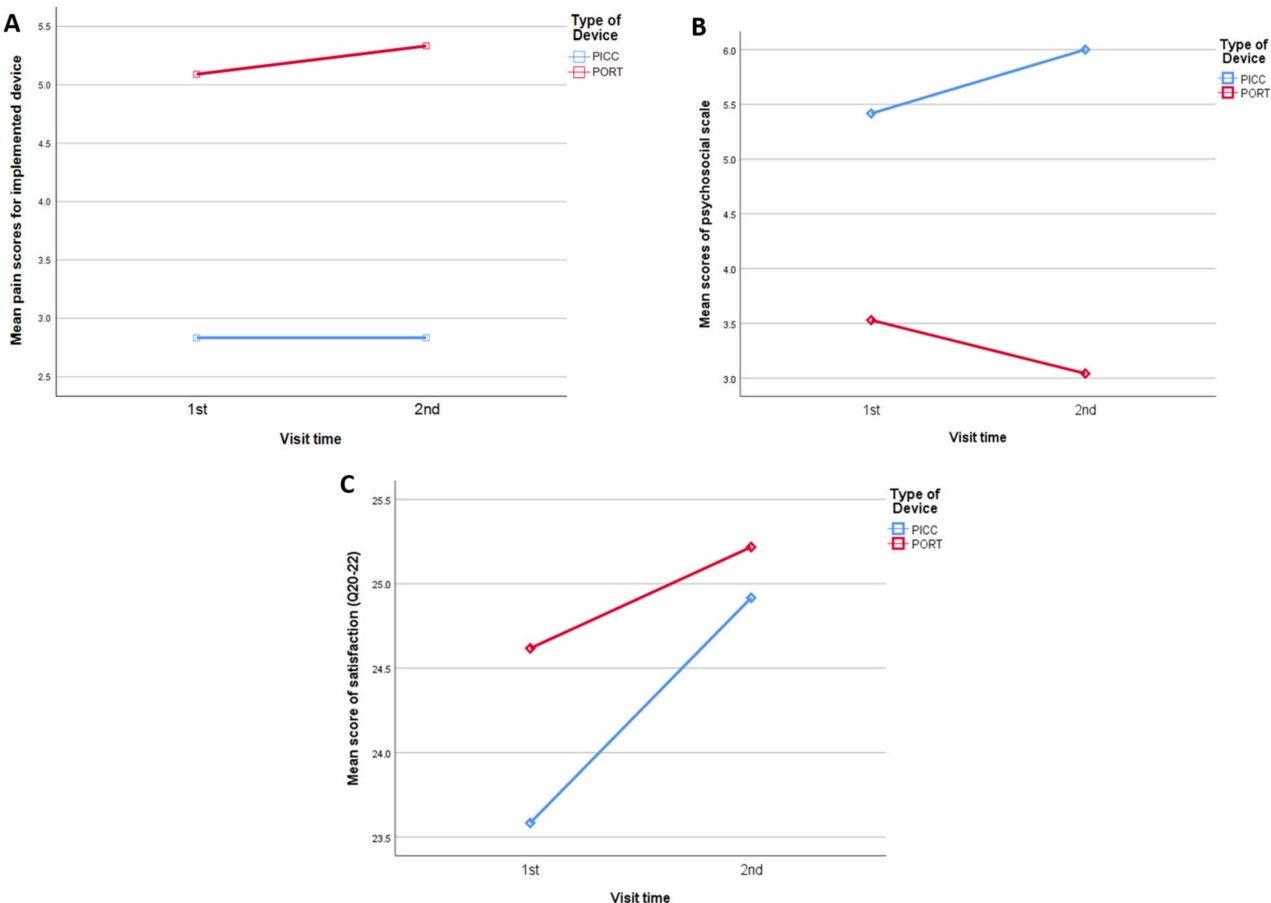

**Figure 1.** Comparison of QLVAD scores between patients with port and PICC devices: (**A**) patients with a port had high pain scores compared with patients with a PICC line; (**B**) patients with a PICC line experienced higher negative psychosocial scores; (**C**) both patients with a PICC and port had high satisfaction scores with their devices at 0 and 3 months.

Table 3 describes several characteristics related to the psychological aspect for the port and PICC line groups at one month and three months. Overall, a significantly higher proportion of subjects with a PICC versus a port reported changes in the way they dressed, difficulty with showering or bathing, and having people comment on the presence of their device in the survey performed at baseline and at three months. At three months, 66.7% of subjects with a PICC versus 33.3% of subjects with a port felt they had changed the way they dressed due to their device (OR = 4.0, 95% CI: 1.2–13.3, $p$ = 0.02). Overall, 88.2% subjects in the PICC group compared to 18.3% in the port group reported difficulties with showering, bathing, or performing personal hygiene activities due to their device (OR = 18.3, 95% CI: 3.5–97.1, $p$ < 0.0001). Similarly, 41.7% of subjects with a PICC compared to 12.5% of subjects with a port experienced comments from people about their device (OR = 5.0, 95% CI: 1.2–21.5, $p$ = 0.02). At three months, 45.8% of patients with a PICC worried that their device may become infected compared to 8.3% of subjects with a port (OR = 9.3, 95% CI: 1.8–48.7, $p$ = 0.003). No significant differences were noted between the two groups regarding sports, exercise, social activities, or the degree of discomfort in between treatments (Table 3).

When QLAVD survey questionnaires were examined by aggregating them into procedural, psychosocial, treatment, and satisfaction constructs, as reported in the Methods section, the results showed that the PICC procedure was significantly less painful than the port but was associated with a significantly higher level of negative psychosocial scores (Table 4). For example, the three-month mean procedural score for the PICC device was 2.8 (±2.3) compared to 5.3 ± (2.9) for the port ($p$ = 0.001), but the mean score for a negative

effect of the PICC device on psychosocial function was 6.0 ($\pm$4.1) compared to 3.0 ($\pm$2.7) for the port ($p$ = 0.005). No significant differences were noted between the two groups in relation to treatment construct and satisfaction with the device.

**Table 3.** Proportions, odds ratios, and 95% confidence intervals of various QLVAD scales at baseline and 3 months between the PICC line and the Port groups.

| QLVAD Constructs | 1 Month % * | | OR (95% CI) *p*-Values | 3 Months % * | | OR (95% CI) *p*-Values |
|---|---|---|---|---|---|---|
| | PICC | Port | | PICC | Port | |
| Do you find it easy to present your device for treatment or blood sampling? | 97.2 | 82.4 | 7.5 (0.85–66.0) 0.052 | 100 | 91.7 | 1.1 (0.97–1.23) 0.49 |
| Do you feel that your device is too visible? | 19.4 | 11.8 | 1.8 (0.47–6.8) 0.37 | 37.5 | 16.7 | 3.0 (0.77–11.6) 0.10 |
| Do you feel that your device is unsightly or ugly? | 27.8 | 11.8 | 2.9 (0.81–10.3) 0.09 | 41.7 | 16.7 | 3.6 (0.93–13.7) 0.05 |
| Do you feel that you have changed the way you dress due to your device? | 61.1 | 17.6 | 7.3 (2.4–22.2) <0.001 | 66.7 | 33.3 | 4.0 (1.2–13.3) 0.02 |
| Have people commented on your device when they see it? | 33.3 | 14.7 | 2.9 (0.89–9.4) 0.07 | 41.7 | 12.5 | 5.0 (1.2–21.5) 0.02 |
| Do you try to cover your device with clothing? | 55.6 | 38.2 | 2.02 (0.19–1.3) 0.15 | 62.5 | 37.5 | 2.80 (0.11–1.3) 0.08 |
| Does your device make you feel anxious (worried)? | 19.4 | 8.8 | 2.50 (0.60–10.6) 0.30 | 16.7 | 4.2 | 4.6 (0.47–44.6) 0.34 |
| Are you worried that your device might become damaged? | 50.0 | 29.4 | 2.4 (0.90–6.4) 0.07 | 41.7 | 29.2 | 1.7 (0.52–5.7) 0.36 |
| Are you worried that your device might become blocked? | 44.4 | 32.4 | 1.7 (0.63–4.4)0.30 | 45.8 | 33.3 | 1.7 (0.52–5.4) 0.37 |
| Are you worried that your device might become infected? | 58.3 | 35.3 | 2.6 (0.97–6.7) 0.054 | 45.8 | 8.3 | 9.3 (1.8–48.7) 0.003 |
| Does your device bother you when performing your work-related activities? | 25.0 | 26.5 | 0.92 (0.31–2.7) 0.88 | 33.3 | 12.5 | 3.5 (0.80–15.3) 0.08 |
| Does your device bother you when you shower, bathe, or perform personal hygiene? | 75.9 | 24.1 | 6.1 (2.1–17.6) 0.001 | 88.2 | 11.8 | 18.3 (3.5–97.1) <0.0001 |
| Does your device bother you when you engage in sports or exercise? | 38.9 | 17.6 | 3.0 (1.0–9.0) 0.049 | 29.2 | 25.0 | 1.2 (0.34–4.4) 0.74 |
| Does your device bother you during social activities? | 16.7 | 11.8 | 1.5 (0.38–5.9) 0.55 | 12.5 | 4.2 | 3.2 (0.32–34.0) 0.60 |
| Does your device bother you when you are lying down in bed? | 50.0 | 35.3 | 1.8 (0.70–4.8) 0.21 | 41.7 | 33.3 | 1.4 (0.44–4.6) 0.55 |
| Does your device hurt? | 36.1 | 32.4 | 1.2 (0.44–3.2) 0.74 | 20.8 | 20.8 | 1.0 (0.24–4.0) 0.999 |
| Is device uncomfortable to touch? | 11.1 | 20.6 | 0.48 (0.13–1.8) 0.28 | 8.3 | 16.7 | 0.44 (0.08–2.8) 0.66 |

* Proportional to patients who answered "yes" to QLVAD questions. Table 3 includes individual questions from the QLVA questionnaire which have yes/no answering options. For the following questions, fewer than 5 patients responded, therefore, data are not presented: Does your device make you feel angry? Does your device make you feel self-consciousness? Does your device remind you of your illness? Does your device make you feel embarrassed?

**Table 4.** Mean QLVAD scale and standard deviation (SD) at baseline and 3 months between the PICC line and the Port groups.

| QoL Constructs | 1 Month | | *p*-Value | 3 Months | | *p*-Value |
|---|---|---|---|---|---|---|
| | PICC | Port | | PICC | Port | |
| Procedural scale [a] (pain and discomfort) | 2.8 ± 2.2 | 5.1 ± 2.9 | 0.001 | 2.8 ± 2.3 | 5.3 ± 2.9 | 0.001 |
| Psychosocial scale [b] (device causing trouble) | 5.4 ± 3.1 | 3.5 ± 3.3 | 0.016 | 6.0 ± 4.1 | 3.0 ± 2.7 | 0.005 |
| Treatment scale [c] | 2.2 ± 2.1 | 3.0 ± 2.1 | 0.13 | 2.1 ± 1.9 | 2.6 ± 2.0 | 0.39 |
| Satisfaction with device [d] | 23.6 ± 5.9 | 24.6 ± 7.1 | 0.51 | 25.0 ± 6.6 | 25.2 ± 5.9 | 0.87 |

Table 4 compares mean scores of different constructs/scales developed from QLVAD for two devices. [a] question related to pain and discomfort during device insertion at 1 to 10 scale: higher score suggests more discomfort; [b] sum of 18 related questions: higher score suggests more trouble with the device; [c] sum of 8 related questions: higher score suggests a negative effect; [d] sum of 3 related questions: higher score suggests more satisfaction.

The mixed effects model examined changes in mean scores for different constructs between the PICC and port groups from baseline to three months after adjusting for other variables (Table S2). The interaction terms and other covariates were not significant; therefore, only a model with the type of device and survey time points is presented here. The results showed that subjects with a PICC had significantly lower pain score estimates when it was accessed by needle for treatment purposes compared to subjects with a port (β = −1.98, 95% CI: −0.92 to −3.05, *p* < 0.001). Survey time variable was not significant; therefore, there was no change in the mean pain score for both devices at three months. Conversely, subjects with a port had a significantly better psychosocial score estimate than patients with a PICC line (β = 2.18, 95% CI: 0.83 to 3.53, *p* = 0.002). However, there was no change in the mean psychological score for both devices at three months. Results for the QLC-30 survey did not reveal any statistically significant changes in mean scores for the different scales between the surveys conducted at baseline and three months for the devices investigated.

*3.3. Complications*

Complications detected by auditing the PACS are compiled in Table S3 Venous thrombosis was the most common complication and was related to arm symptoms for most subjects, except for one subject who only had chest pain. No significant difference in the complication rate was noted between the two groups. Overall, four (8%) subjects with a PICC compared to six (12%) subjects with a port developed occlusive deep venous thrombosis (DVT). Four of 15 (27%) subjects with the Bard port versus two of 36 (5%) subjects with the Cook port developed DVT (*p* = 0.05).

**4. Discussion**

It is understandable that subjects with the port experience more pain at the time of device insertion and during chemotherapy infusion. Port insertion requires a skin incision and the creation of a subcutaneous pocket in which the device resides. Skin closure is also required to facilitate wound healing. It is a minor surgical procedure with risks of hemorrhage, infection, and pain.

Infusion of chemotherapy for those with a port requires the direct insertion of a non-coring infusion needle through the skin into the inner chamber. The pain is similar to that involved for phlebotomy. However, accessing the port is much more predictable and reliable in the hands of trained personnel, resulting in fewer needle punctures than for those patients who do not have a port or PICC.

Given the findings associated with port device pain, this could be considered an important issue to raise with patients when discussing the type of venous access device required for infusion therapy. If patients have a severe needle phobia, a PICC may be more appropriate.

Ports had very little impact on how the subjects dress, because they are associated with a small upper arm incision and a small protrusion of the skin on the upper arm. There are no components of the port that hang from the arm or are external to the skin. Hence, there is no need to wrap or fixate the port. The PICC, on the other hand, requires great care in covering the device with a clean dressing and a fixation device that protects the PICC from inadvertent extraction and infection. Additionally, some patients may choose to dress in a fashion that protects the PICC from inadvertent damage by it becoming caught on something in one's surroundings and becoming inadvertently extracted. Clothing choices are definitely not as limited with the port as with the PICC.

Difficulty with showering or bathing with the PICC is related to the need to keep the PICC insertion site clean and dry. This is not an issue with a healed port incision because patients are able to swim, bathe, shower, and participate in unrestricted water activities once the insertion site incision has healed. If patients are avid water activity enthusiasts, this is an important differential functionality between the two devices. Overall, the level of activity related to work, sports, and social interactions should also be taken into consideration.

Having people negatively comment on the presence of the device is a result of the fixation of the device, the site dressing, and the external limbs with the PICC being readily visible, which is not the case for the port due to its subcutaneous location. This is unavoidable due to the differences in device design. Usually, the only way to avoid this is to dress in a fashion that covers the implantation site and prevent others from seeing the location of the device. The mixed effects model did not show significant changes in mean scores for different constructs including mean pain and psychological scores (improvement or worsening) between the PICC and port groups from baseline to three months. This suggests that patients with a PICC line continue to experience less pain during device access, but a high negative psychological effect compared to subjects with a port.

The complication profiles for subjects with a PICC or port seem comparable, with the exception that the power injectable port does seem to be associated with a slightly higher rate of venous thrombosis, although this was not found to be statistically significant in this cohort. Additionally, the detection of venous thrombosis was based on clinical signs and symptoms; this may not be a reliable method of thrombosis detection because there is a tendency for asymptomatic venous thrombosis to develop in this patient group [16]. Given that power injectable venous devices are usually associated with a larger catheter diameter, which has been proven to increase the risk of venous thrombosis, the role that power injection may play in the treatment and follow-up of patients should be balanced against the potential risk for venous thrombosis [5].

Although our study showed that both groups had similar complication rates, other studies have shown that PICCs compared to ports tend to be associated with a high complication rate [17–21]. For example, Patel et al. prospectively examined CVC-related complications in 70 patients with a solid tumor and reported that, overall, 6% of the patients with a port compared to 20% patients with a PICC line experienced major complications [17]. A systematic review of the literature showed that PICC compared to the other central venous lines including a port was associated with an increased risk for DVT [20]. In the present study, the equivalent complication rates related to the device could be a random occurrence. Furthermore, complications leading to PICC removal may have been underestimated if the device was removed and the reason for removal was not recorded in the medical record. The rate of non-occlusive and occlusive venous thrombosis detected using ultrasound was outside the expected norm, which may very well be related to catheter material and catheter size. When the project was initiated, a 6F power port was used for arm implantation and power injection. We have transitioned to a smaller catheter diameter with a different manufacturer. Finally, there are risks of contrast extravasation with simple peripheral intravenous catheters, PICCs and ports [22,23]. The ports are designed and rated for high-volume contrast, being injected under increased pressure. Contrast extravasation is a nuisance complication that fortunately rarely occurs and usually

does not have dire consequences, especially now that contrast is low-osmolality and non-ionic in chemical composition. It is important that ports must be properly accessed by trained healthcare professionals.

Long-term comparison of these two different venous access devices is limited by the expectation that PICCs will be employed for intravenous treatment for a shorter time frame, i.e., usually less than six months. This results in a significant drop-out effect for data capture for those subjects with a PICC that may have confounded complication rates in favor of PICC lines. In addition, it is much easier to have a PICC removed, and clinicians are more inclined to simply remove a PICC and have a new one implanted if there are any potential complications, regardless of the severity of the complication. Hence, one of the limitations of our study is the lack of longer-term follow-up and detailed information on duration and reasons of the removal of PICC lines.

An additional limitation of our pilot study is the small cohort size. However, the cohort was found to be statistically comparable to the EORTC QOL survey results, suggesting that there were no substantial differences in the overall group make-up that would adversely affect the results of the QLAVD survey. The groups were balanced, with 50 subjects in the PICC group and 51 subjects in the port group. However, no attempt was made to balance the group according to the stage of their malignancy. This could be a potential factor to consider for future research. Our results do not suggest that quality of life was diminished between the two major groups, i.e., PICC vs. port. Nevertheless, we do not know if the frequency of line care, such as venous catheter flushing, has adverse effects on the QoL of patients with a venous catheter who are not on active treatment. Lastly, this study was performed in subjects with breast and colorectal cancers, and findings may not be generalizable to patients with other solid organ and hematological cancers.

In conclusion, this pilot study suggests that the decision to employ a venous access device for the treatment of a malignancy is a multi-factorial one. The duration of treatment cannot be the only factor taken into consideration. Determination of the degree of needle phobia, levels of sporting and social activities, ability of the patient to swim, shower, and bathe, and the need to alter one's dressing habits should also be considered. The need for a power-injectable device is also potentially clouded by the possibility of an increased risk of venous thrombosis, which should be taken into account.

**Supplementary Materials:** The following are available online at https://www.mdpi.com/article/10.3390/curroncol28020141/s1 Table S1 Quality of Life Questionnaires related to the venous access device (QLVAD), Table S2 Mixed effects model showing QoL of patients with a port versus a PICC, Table S3: PACS detected complications related to arm PICCs and Port.

**Author Contributions:** Conceptualization, B.B., H.L. (Hyun Lim) and S.A.; Data curation, B.B., T.A., A.S. and S.A.; Formal analysis, H.L. (Hyun Lim) and H.L. (Ha Le); Investigation, B.B., L.D., T.A., A.S. and S.A.; Methodology, B.B., H.L. (Hyun Lim) and S.A.; Project administration, B.B., L.D. and S.A.; Software, H.L. (Hyun Lim) and H.L. (Ha Le); Supervision, B.B. and S.A.; Writing—original draft, B.B.; Writing—review & editing, B.B., H.L. (Hyun Lim), L.D., H.L. (Ha Le), T.A., A.S. and S.A. All authors have read and agreed to the published version of the manuscript.

**Funding:** The research was funded and was supported by the clinical trial unit.

**Institutional Review Board Statement:** The study was approved by the University of Saskatchewan Ethics Board "Bio-15-248 Arm Vein Port (TIVAD) vs. PICC for the Treatment of Breast and Colon Carcinoma: A Pilot Study. The University of Saskatchewan Biomedical Research Ethics Board (Bio-REB) has reviewed the above-named research study. The study was found to be acceptable on scientific and ethical grounds. The principal investigator has the responsibility for any other administrative or regulatory approvals that may pertain to this research study, and for ensuring that the authorized research is carried out according to governing law. This approval is valid for the specified period provided there is no change to the approved protocol or consent process.

**Informed Consent Statement:** All procedures performed in studies involving human participants were in accordance with the ethical standards of the institutional and/or national research committee and with the 1964 Helsinki declaration and its later amendments or comparable ethical standards. The University of Saskatchewan's Biomedical Research Ethic Board approved the study and informed consent was taken from all the participants.

**Data Availability Statement:** The data presented in this study may available after ethics an operation approval from Data Access Committee on request from the corresponding author. The data are not publicly available due to ethics requirement and institutional policy.

**Acknowledgments:** We are thankful for the support by the Clinical Trial Unit at the Saskatchewan Cancer Center (SCA). We are also thankful to Data Access committee at SCA and the Saskatchewan Health Authority (former Saskatoon Health Region) for operational approval, and Jennifer Simmons for administrative assistance. Part of the data were presented in the 2020 San Antonio Breast Cancer Conference.

**Conflicts of Interest:** No authors have a conflict of interest to disclose.

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
