# Peer review of "Comparison of the Quality of Life of Patients with Breast or Colon Cancer with an Arm Vein Port (TIVAD) Versus a Peripherally Inserted Central Catheter (PICC)"

_curroncol, doi:10.3390/curroncol28020141_

Round 1

Reviewer 1 Report

To the authors,

I congratulate the authors, as this paper represents a huge work of qol questionnaire analysis and statistical assessment.

The paper is of  meaningful interest  that deserves to be published. This is well built, and displayed.

However  some modifications could be done:

1-Material & Methods chapter:  study population table is lacking : mean age (range) , dominant arm, BMI, side of insertion of PICC/ arm port (whenever possible).

2-Results: very well presented as qol questionnaires can be very time consuming and not that easy to discuss. Please add the  complication rates in percentages and also in /100patient days of implantation (this is the way of presentation in most of venous catheter complications presentations).

3-Discussion: the paper deals with the question: "venous catheter PICC or arm port devices in breast/colon oncology patients :  does it matter ?" 

The authors gave their conclusions according to the results of their study. May I suggest to add the following references to be discussed and added  in this chapter ?

Patel GS,et al . Comparison of peripherally inserted central venous catheters (PICC) versus subcutaneously implanted port-chamber catheters by complication and cost for patients receiving chemotherapy for non-haematological malignancies. Support Care Cancer. 2014 Jan;22(1):121-8. doi: 10.1007/s00520-013-1941-1. Epub 2013 Sep 5. PMID: 24005884.

Qi F, Cheng H, Yuan X, Zhang L. Comparison of PICC and TIVAP in chemotherapy for patients with thyroid cancer. Oncol Lett. 2020 Aug;20(2):1657-1662. doi: 10.3892/ol.2020.11732. Epub 2020 Jun 15. PMID: 32724407; PMCID: PMC7377162.

Marcy PY, et al. Radiological and surgical placement of port devices: a 4-year institutional analysis of procedure performance, quality of life and cost in breast cancer patients. Breast Cancer Res Treat. 2005 Jul;92(1):61-7. doi: 10.1007/s10549-005-1711-y. PMID: 15980992.

Johansson E, Hammarskjöld F, Lundberg D, Arnlind MH. Advantages and disadvantages of peripherally inserted central venous catheters (PICC) compared to other central venous lines: a systematic review of the literature. Acta Oncol. 2013 Jun;52(5):886-92. doi: 10.3109/0284186X.2013.773072. Epub 2013 Mar 11. PMID: 23472835.

Moureau N, Chopra V. Indications for peripheral, midline and central catheters: summary of the MAGIC recommendations. Br J Nurs. 2016 Apr 28-May 11;25(8):S15-24. doi: 10.12968/bjon.2016.25.8.S15. PMID: 27126759.

I would also build some  hypotheses regarding the evolution of Qol scores between the first month and the third month of implantation  between the two techniques arm ports and piccs in Table 2, in the discussion chapter.

Discuss why the complication rates are equivalent in both groups while all other papers show fewer complications in ports versus Picc lines.

Regarding power Piccs and Power ports, In my opinion, I am scared about extravasation or occlusion risks rather than by venous thrombosis.

I would briefly discuss the following references:

Schummer C, Sakr Y, Steenbeck J, Gugel M, Reinhart K, Schummer W. Risk of extravasation after power injection of contrast media via the proximal port of multilumen central venous catheters: case report and review of the literature. Rofo. 2010 Jan;182(1):14-9. doi: 10.1055/s-0028-1109742. Epub 2009 Oct 26. PMID: 19859861.

Marcy PY, Thariat J, Figl A. Power injection via a venous port: a new challenge for radiologists. Rofo. 2010 Jun;182(6):536; author reply 536-7. doi: 10.1055/s-0029-1245439. Epub 2010 Jun 1. PMID: 20517796.

Conclusion: 

The authors' conclusions are very useful as they give the reader a pragmatic outlook to the operator community:

The decision for Picc or arm port is a multi-factorial one, depending on duration time (first factor to my opinion), needle phobia (second factor), levels of sporting and social activities (third factor) to be considered.

The need for a power-injectable device is also potentially clouded by the possibility of an increased risk of complication (occlusion, extravasation, thrombosis) , which should be taken into account.

Author Response

Thank you very much for your correspondence about the above manuscript submitted for consideration for publication in Current Oncology. We appreciated the comments made by the reviewers and have thoroughly addressed them in the second revision of the manuscript as described below:

Reviewer 1

I congratulate the authors, as this paper represents a huge work of QOL questionnaire analyses for access to Cancer Agency chart.  The paper is of meaningful interest that deserves to be published. This is well built, and displayed.

Thanks very much for a very positive feedback, we appreciate it very much.

However some modifications could be done:

1-Material & Methods chapter:  study population table is lacking: mean age (range), dominant arm, BMI, side of insertion of PICC/ arm port (whenever possible).

We have collected limited information on patients’ demographic, disease status and treatment, therefore a table is not provided. However, information of mean age (range) and dominant arm has been included in the results section (lines162-163). With respect to sidedness of the device we have provided information on the methods section in lines 83-86. We have not collected information on BMI.   

2-Results: very well presented as QOL questionnaires can be very time consuming and not that easy to discuss. Please add the complication rates in percentages and also in /100patient days of implantation (this is the way of presentation in most of venous catheter complications presentations).

Complication rates in % and days from a catheter insertion to a complication are provided in parentheses in table 6 (supplement).  The major goal of our project was to assess quality of life.  Due to drop out we do not have detailed information on the duration of device in all patients. This data is especially not available for PICCs as many were removed in the outpatient oncology care center, physician’s office, on the ward, or in other clinical locations in patients home towns.  This limitation is highlighted in discussion in line 339-340 and 335-336. 

3-Discussion: the paper deals with the question: "venous catheter PICC or arm port devices in breast/colon oncology patients:  does it matter ?" 

We appreciate this comment. The study was done in patients with breast or colorectal cancer, two most common groups of patients with solid tumor who require a central venous access for systemic therapy.  We do not know if the study findings can be generalized to patients with other solid organ and hematological malignancies. We have highlighted it in the discussion in lines 350-352.  

4-The authors gave their conclusions according to the results of their study. May I suggest to add the following references to be discussed and added in this chapter?

Patel GS,et al . Comparison of peripherally inserted central venous catheters (PICC) versus subcutaneously implanted port-chamber catheters by complication and cost for patients receiving chemotherapy for non-haematological malignancies. Support Care Cancer. 2014 Jan;22(1):121-8. doi: 10.1007/s00520-013-1941-1. Epub 2013 Sep 5. PMID: 24005884.

Qi F, Cheng H, Yuan X, Zhang L. Comparison of PICC and TIVAP in chemotherapy for patients with thyroid cancer. Oncol Lett. 2020 Aug;20(2):1657-1662. doi: 10.3892/ol.2020.11732. Epub 2020 Jun 15. PMID: 32724407; PMCID: PMC7377162.

Marcy PY, et al. Radiological and surgical placement of port devices: a 4-year institutional analysis of procedure performance, quality of life and cost in breast cancer patients. Breast Cancer Res Treat. 2005 Jul;92(1):61-7. doi: 10.1007/s10549-005-1711-y. PMID: 15980992.

Johansson E, Hammarskjöld F, Lundberg D, Arnlind MH. Advantages and disadvantages of peripherally inserted central venous catheters (PICC) compared to other central venous lines: a systematic review of the literature. Acta Oncol. 2013 Jun;52(5):886-92. doi: 10.3109/0284186X.2013.773072. Epub 2013 Mar 11. PMID: 23472835.

Moureau N, Chopra V. Indications for peripheral, midline and central catheters: summary of the MAGIC recommendations. Br J Nurs. 2016 Apr 28-May 11;25(8):S15-24. doi: 10.12968/bjon.2016.25.8.S15. PMID: 27126759.

As per suggestions we have added most of the references (references 19-22) and the relevant text in discussion (lines 314-320).

I would also build some hypotheses regarding the evolution of QOL scores between the first month and the third month of implantation between the two techniques arm ports and PICCs in Table 2, in the discussion chapter.

The mean scores for most constructs including pain and psychological score did not significantly change at baseline and 3 months (improved or got worse).  This has been highlighted in the revised manuscript in lines 292-296.

Discuss why the complication rates are equivalent in both groups while all other papers show fewer complications in ports versus PICC lines.

We appreciate very much the reviewer comments. We cannot explain why the complication rates were not significantly different between the two devices.  Random occurrence is a key explanation. Additionally, complications leading to PICC removal may have been underestimated if the device was removed and the reason for removal was not recorded in the medical record. We have discussed complication rates associated with the two devices in lines 314 to 323 in the discussion section of the revised manuscript and have provided other references (reference 19-23) as mentioned above.  

Regarding power PICCs and Power ports, in my opinion, I am scared about extravasation or occlusion risks rather than by venous thrombosis.

I would briefly discuss the following references:

Schummer C, Sakr Y, Steenbeck J, Gugel M, Reinhart K, Schummer W. Risk of extravasation after power injection of contrast media via the proximal port of multilumen central venous catheters: case report and review of the literature. Rofo. 2010 Jan;182(1):14-9. doi: 10.1055/s-0028-1109742. Epub 2009 Oct 26. PMID: 19859861.

Marcy PY, Thariat J, Figl A. Power injection via a venous port: a new challenge for radiologists. Rofo. 2010 Jun;182(6):536; author reply 536-7. doi: 10.1055/s-0029-1245439. Epub 2010 Jun 1. PMID: 20517796.

Thanks for an important comment. We have added the two references (references 24-25) and relevant text about contrast extravasation in the discussion section in lines 326-331.

Conclusion: 

The authors' conclusions are very useful as they give the reader a pragmatic outlook to the operator community:

The decision for PICC or arm port is a multi-factorial one, depending on duration time (first factor to my opinion), needle phobia (second factor), and levels of sporting and social activities (third factor) to be considered.

The need for a power-injectable device is also potentially clouded by the possibility of an increased risk of complication (occlusion, extravasation, thrombosis), which should be taken into account.

Thanks very much for this comment.

Reviewer 2 Report

Interesting paper requiring revision :

1) Line 238 lower pain for PICCs ; please add and clarify that you are referring to pain only when accesed by needle puncture for therapy delivery.

2) Line 250. Comment on occlusive dvt possibly related to catheter size ( Bard power was a 6 F)has been well addressed on the discussions ection. However it is unclear in the materials and methods section if power 6f were inserted peripherally or by a cenytral route; Authors must disclose this point.

3) It is in some manner questionable the decision of using a 6F port power   instead of other of the available 5F PUR power ports devices implatble even at the upper arm. This point must be addressed as a study limitation in the discussion section.

4)Overall complications as described by the Authors appears out of range as compared to literature data especially for arm ports, see recent published studies of Taxbro at al, - Tippit et al – Clatot et al- Bertoglio et al. It seems in some manner, as data are presented ,that your devices were poor in terms of efficacy and safety or that something is wrong in the institutional protocols for implant and nursing. Can you please comment on that?

5)On the basis of what expressed in point 4 it would be advisable to disclose also the exactly rates of removal of the devices for complications.

6) Provided that the association of QLQ-C30 and modified version of QASICC appears to well fit with the primary outcome of the study, please consider that you have not evaluated an important questionable aspect of PICCs chronic use that is related to the need of continuous nursing of the device. In general literature and guidelines suggest a weekly to 15 days maintenance for PICCs. In terms of QoL this should have been considered in the study design or disclosed as a study limitation.

Author Response

Reviewer 2

1) Line 238 lower pain for PICCs; please add and clarify that you are referring to pain only when accessed by needle puncture for therapy delivery.

 It has been clarified in line 246-47 of the revised manuscript.

2) Line 250. Comment on occlusive DVT possibly related to catheter size (Bard power was a 6 F) has been well addressed on the discussions section. However it is unclear in the materials and methods section if power 6f were inserted peripherally or by a central route; Authors must disclose this point.

It has been specified in lines 82-83 that all venous access devices were implanted into the arm, between the antecubital fossa and the axilla, using ultrasound or arm venography for venous access. 

3) It is in some manner questionable the decision of using a 6F port power instead of other of the available 5F PUR power ports devices implantable even at the upper arm. This point must be addressed as a study limitation in the discussion section.

 We agree with the reviewer comments. However, when the project was initiated this was the port identified that provided an opportunity for arm implantation and power injection.  We have since transitioned to a smaller catheter diameter with a different manufacturer.  We have clarified it in discussion section in lines 323-326.

4) Overall complications as described by the Authors appears out of range as compared to literature data especially for arm ports, see recent published studies of Taxbro at al, - Tippit et al – Clatot et al- Bertoglio et al. It seems in some manner, as data are presented that your devices were poor in terms of efficacy and safety or that something is wrong in the institutional protocols for implant and nursing. Can you please comment on that?

 We work in a tertiary care institution with established standards of care that are recognized nationally and internationally.  We acknowledge that the rate of non-occlusive and occlusive US detected venous thrombosis is outside the expected norm and may very well be related to the catheter material and the catheter size.  However, complications described in this manuscript are not meant to be used for a full analysis of the merits of each device being used but rather are provided to develop the context for the quality-of-life analysis performed with the general concept that increased complications may well result in diminished quality of life for a specific device.  Our data does not suggest that quality of life was diminished between the two major groups i.e. PICC vs. port. We have highlighted it in the discussion of the revised manuscript in lines 314-326.

5) On the basis of what expressed in point 4 it would be advisable to disclose also the exactly rates of removal of the devices for complications.

The complete data of device removal for complications is only available in patients with ports. Three ports were prematurely removed for a complication – 1 for wound dehiscence, 1 for device occlusion, and 1 for infection/septicemia.  Since we do not have detailed information about the duration and reasons for the removal of device in all patients with PICCs we have not reported it. This limitation is highlighted in discussion in lines 339-340.

6) Provided that the association of QLQ-C30 and modified version of QASICC appears to well fit with the primary outcome of the study, please consider that you have not evaluated an important questionable aspect of PICCs chronic use that is related to the need of continuous nursing of the device. In general literature and guidelines suggest a weekly to 15 days maintenance for PICCs. In terms of QoL this should have been considered in the study design or disclosed as a study limitation.

Thanks for the comments. It is clarified in the method section in lines 94-95 that as per institutional protocol PICCs were flushed once weekly whereas ports were flushed every 4 week when not in use. In the present study most patients were on active treatment that preclude additional visits for venous catheter flushing. We have highlighted it in study limitation in lines 347-350.

Reviewer 3 Report

General Comments

In their paper entitled “Comparison of the Quality of Life of Patients with Breast or Colon Cancer with an Arm Vein Port (TIVAD) versus a Peripherally Inserted Central Catheter (PICC)” authors present a 1 year prospective cohort study to assess Quality of Life (QOL) of patients with colorectal or breast cancer who receive CVC. Secondary aims were to compare satisfaction and QOL and complication rates of patients with ports compared to patients with PICCs.

Specific Comments

Title: No comment.

Abstract: “...were administered...over 1-year period.” From January 2016 to December 2018 is more like a 3-year period.

Keywords: Might as well add the word “complications”.

  • In order to answer the first question of the aim on CVC satisfaction among cancer patients one has to compare results also with a group that used no CVC.
  • The many drop-outs beyond the 3rd month may present a confounder in favour of the PICC-arm. This should be mentioned in the discussion section.
  • The cohort is large enough in every arm, but suffers from lack of adjustment of the results to important confounders such as:
  1. Age and BMI
  2. Diameter and type of vein used for access. In how many cases did the authors use the cephalic vein for access?
  3. Use of dominant vs non-dominant arm, or site of breast and/or lymph node operation compared to site of port or PICC placement.
  4. In my opinion the cases where venography was used for access instead of ultrasound should be excluded. The method is obsolete and may be a significant confounder in the assessment of procedural pain, thrombosis rate, etc.

Author Response

3rd Reviewer

General Comments

In their paper entitled “Comparison of the Quality of Life of Patients with Breast or Colon Cancer with an Arm Vein Port (TIVAD) versus a Peripherally Inserted Central Catheter (PICC)” authors present a 1 year prospective cohort study to assess Quality of Life (QOL) of patients with colorectal or breast cancer who receive CVC. Secondary aims were to compare satisfaction and QOL and complication rates of patients with ports compared to patients with PICCs.

Specific Comments

Title: No comment.

Abstract: “...were administered...over 1-year period.” From January 2016 to December 2018 is more like a 3-year period.

Thanks for the comment. We have clarified in the abstract that study recruitment period was 3 years and QOL questionnaires was recorded up to 1 year after the insertion of a venous catheter. 

Keywords: Might as well add the word “complications”.

We have added complications in keywords.

  • In order to answer the first question of the aim on CVC satisfaction among cancer patients one has to compare results also with a group that used no CVC.

We appreciate very much this comment however, Quality of Life Assessment Venous Device (QLAVD) survey was designed for patients with a venous access device. Hence we do not have a control group with no venous access device.

  • The many drop-outs beyond the 3rdmonth may present a confounder in favor of the PICC-arm. This should be mentioned in the discussion section.

We have added it in the discussion in lines 335-336

  • The cohort is large enough in every arm, but suffers from lack of adjustment of the results to important confounders such as:
  1. Age and BMI

We did not collect information on BMI. All the other variables including sex and dominant arm were not correlated with QOL.

  1. Diameter and type of vein used for access. In how many cases did the authors use the cephalic vein for access?

We did not collect data on vein size and type of vein. The veins selected for catheter insertion were preferentially the basilic vein, then the brachial vein and followed by the cephalic vein.  It has been clarified in the lines 83-84 of the revised manuscript

  1. Use of dominant vs non-dominant arm, or site of breast and/or lymph node operation compared to site of port or PICC placement.

The use of dominant vs non-dominant arm in mixed effect model was not associated QOL. It has been specified in the revised manuscript in lines 83-86 that the preferential arm for device insertion for trial subjects was their non-dominant arm. For women with breast cancer contralateral arm was used for device insertion.

  1. In my opinion the cases where venography was used for access instead of ultrasound should be excluded. The method is obsolete and may be a significant confounder in the assessment of procedural pain, thrombosis rate, etc

We have for the most part converted to US guided venous access and only few patients required venography.  It has been specified in lines 168-169

We sincerely thank the reviewers for their helpful comments. We hope that the changes we have made meet the expectations of the reviewers and that this work is now suitable for publication.

Round 2

Reviewer 3 Report

The paper in its present form meets the standards for publication.